# Reduced Taurine Serum Levels in Inflammatory Bowel Disease

**DOI:** 10.3390/nu16111593

**Published:** 2024-05-23

**Authors:** Rachele Frascatani, Adelaide Mattogno, Andrea Iannucci, Irene Marafini, Giovanni Monteleone

**Affiliations:** 1Department of Systems Medicine, University of Rome “Tor Vergata”, 00133 Rome, Italy; rachele.frascatani@gmail.com; 2Policlinico Universitario Tor Vergata, 00133 Rome, Italyirene.marafini@ptvonline.it (I.M.); 3Department of Biomedicine and Prevention, University of Rome “Tor Vergata”, 00133 Rome, Italy; andreaiannucci93@gmail.com

**Keywords:** Crohn’s disease, ulcerative colitis, IBD, amino acids

## Abstract

Taurine is a semi-essential micronutrient that acts as an anti-inflammatory molecule. The oral administration of taurine to colitic mice attenuates ongoing mucosal inflammation. This study aimed to determine whether inflammatory bowel diseases (IBDs) are marked by changes in the circulating levels of taurine. We measured the serum concentrations of taurine in 92 IBD patients [46 with ulcerative colitis (UC) and 46 with Crohn’s disease (CD)] and 33 healthy controls with a commercial ELISA kit. The taurine levels were significantly decreased in both patients with UC and patients with CD compared to the controls, while there was no difference between CD and UC. Taurine levels declined with age in healthy controls but not in IBDs. IBD patients younger than 50 years had levels of taurine reduced compared to their age-matched controls. In the IBD group, taurine levels were not influenced by the body mass index of the patients and the consumption of taurine-rich nutrients, while they were significantly reduced in UC patients with clinically active disease compared to those in clinical remission. These findings indicate that IBDs are marked by serum taurine deficiency, which would seem to reflect the activity of the disease, at least in UC.

## 1. Introduction

Taurine is a sulfur-containing amino acid that is not incorporated into protein. In mammalian cells, taurine is produced from cysteine through the action of cysteine sulfinic acid decarboxylase, but it can also be derived from the diet and is taken up by cells through taurine transporters [1]. Taurine plays critical roles in the development of some organs (e.g., eye and brain) and physiological functions (e.g., reproduction and osmoregulation) [2,3]. Taurine is abundantly expressed by leucocytes, and accumulating evidence supports its involvement in the control of immune responses, given its ability to inhibit the production of several inflammatory cytokines [4,5]. Consistently, the supplementation of taurine has been shown to attenuate tissue-damaging immunoinflammatory reactions, while taurine deficiency leads to immune activation in several organs [6,7].

Crohn’s disease (CD) and ulcerative colitis (UC) are the major inflammatory bowel diseases (IBDs) in human beings. These pathologies are lifelong disorders characterized by chronic, recurring inflammation in the gastrointestinal tract that can cause various degrees of tissue damage and promote the development of complications [8]. The etiology of both CD and UC remains unknown, but it has been hypothesized that IBD arises in genetically predisposed individuals as a result of the action of multiple environmental factors that trigger an altered mucosal immune response and eventually the pathological process [9]. The rapid increases in the incidence of IBD over the past decades in low-incidence parts of the world (i.e., South America and Asian countries), which have followed the introduction of the Western diet (i.e., high in fat and protein and low in fruits and vegetables) in such countries supports the role of the diet in the onset and/or course of IBD [10]. Consistently, in recent years there has been an enormous effort to identify dietary components, which can either amplify or attenuate IBD-associated mucosal inflammation. In this context, for example, it has been demonstrated that food additives can either directly or indirectly break the intestinal barrier, thus promoting the penetration of luminal microorganisms into the lamina propria with the downstream effect of stimulating detrimental inflammatory responses [11,12]. On the other hand, there is evidence that a diet high in fiber is beneficial to both patients with UC and CD and decreases the incidence of the disease [13]. Studies from various laboratories have also shown that dietary taurine protects intestinal epithelial cell damage, reduces the production of mucosal inflammatory cytokines, and attenuates colitis in experimental murine models of IBD, thus supporting the cytoprotective and anti-inflammatory properties of taurine in the gut [14]. 

This study aimed to determine whether IBDs are marked by changes in serum taurine concentrations.

## 2. Materials and Methods

### 2.1. Study Population and Data Collection

In this observational, prospective study, consecutive adult IBD outpatients were enrolled at a single tertiary IBD center (Tor Vergata University Hospital in Rome, Italy). Patients were examined during routine clinical visits and serum samples were collected from each of them. For each IBD patient, the following demographic and clinical information was collected: type of IBD, current clinical activity, Montreal classification, and concomitant therapy. The extent of IBD was determined using endoscopic examinations and, for CD patients, cross-sectional imaging techniques such as small bowel ultrasound and/or magnetic resonance enterography were carried out either before or at the time of the serum sample collection. Clinical activity was evaluated using the partial Mayo score (PMS) for UC and the Harvey Bradshaw Index (HBI) for CD. Clinically active disease was identified by a PMS ≥ 2 for UC and an HBI ≥ 5 for CD. For each patient, we also collected information relating to the body mass index (BMI) and consumption of foods high in taurine (i.e., seafood and dark meat poultry) and taurine-containing energy drinks. 

Serum samples were also taken from healthy controls who had no clinical history or family history of IBD. All participants provided informed consent to scientific and anonymous use of their clinical data at the time of enrolment into the study database. Individuals under the age of 18 years and those unable to comprehend the informed consent (e.g., due to language barriers) were excluded from the study. The study received approval from the local ethics committee (N.29091/2022).

### 2.2. Enzyme-Linked Immunosorbent Assay

Circulating taurine levels were determined using a commercially available competitive enzyme-linked immunosorbent assay (ELISA) kit (Abbexa, Cambridge, UK) using 50 μL of serum. Absorbance readings were taken at 450 nm using a multimode detector DTX 880 (Beckman Coulter, Milan, Italy).

### 2.3. Statistical Analysis

Qualitative data are presented as numbers and percentages (%), while quantitative data are expressed as the median (range). Differences between groups were assessed using either the Mann–Whitney U test or Student’s *t*-test, depending on the normal distribution of the data. A two-way ANOVA with an interaction term was employed to evaluate the association between serum taurine levels and factors such as the age of the patients/controls, BMI, and consumption of foods and intakes of taurine-containing drinks. All analyses were conducted using Graph-Pad 6 software.

## 3. Results

### 3.1. Patient Characteristics

The study population consisted of 46 patients with UC, 46 patients with CD, and 33 healthy controls (Table 1). The median age was 46 years (range 20–69 years) for UC patients and 46 years (range 19–76 years) for CD patients. Among the UC patients, 25 (54.3%) were female, while among the CD patients, 18 (39.1%) were female. Of the 46 patients with CD, 6 (13%) had lesions confined to the colon, 24 (52.2%) had a disease involving the distal ileum, and 16 (34.8%) had a disease affecting both the colon and ileum. Among the 46 UC patients, 5 (10.9%) had a disease limited to the rectum (E1), 22 (47.8%) had left-sided colitis (E2), and 19 (41.3%) had extensive colitis (E3). Regarding treatment, 23 (50%) CD patients and 32 (69.6%) UC patients were taking mesalamine. Steroids were taken by 4 (8.7%) UC patients and 8 (17.4%) CD patients. Biologics were used by 8 (17.4%) UC patients and 19 (41.3%) CD patients. Additionally, 5 (10.9%) patients with CD were on immunosuppressants. In the control group, the median age was 35.5 years (range 21–69 years), with 24 (72.7%) being females. The UC and CD patient groups were comparable in terms of age and gender. At the time of enrolment, 21 patients with CD (45.6%) and 26 UC patients (56.5%) had active disease. More patients with UC were taking mesalamine (29/46, 63% vs. 15/46, 32.6% in CD, *p* = 0.007), while more CD patients were on biologics (19/46, 41.3% vs. 8/46, 17.4% in UC, *p* = 0.022).

### 3.2. Serum Taurine Levels Are Reduced in IBD

Serum taurine levels were significantly reduced in IBD patients compared to the controls (Figure 1A). Both CD patients and UC patients had taurine levels significantly lower than those in controls, while there was no significant difference between CD and UC. Since the median age of our IBD population was greater than that of the controls (46 vs. 35.5 years) and the circulating levels of taurine decline with age in humans [15], we evaluated the blood taurine concentrations at different ages of IBD and controls by dividing them into three main subgroups. The serum levels of taurine declined with age in healthy controls, and individuals with an age greater than 50 years had levels of taurine significantly reduced as compared to those with age <30 years or >30 and <50 years (Figure 2). In contrast, in IBD, there was no significant decline in taurine levels with age (Figure 2). Notably, the taurine levels were significantly reduced in IBD compared to controls in individuals with age <30 years and in those with age >30 and <50 years (Figure 2). 

### 3.3. Serum Levels of Taurine Are Diminished in IBD Patients with Clinically Active Disease

In IBD, no significant correlation was found between the serum taurine levels and the BMI of the patients (Figure 3A). Taurine levels did not significantly differ between male (median 98.7 ng/mL, range 36–340 ng/mL) and female (median 145.6 ng/mL, range 53–351.85 ng/mL; *p* = 0.9) controls, and between male (median 87 ng/mL, range 21.6–511 ng/mL) and female (median 89 ng/mL, range 23.1–342.2 ng/mL; *p* = 0.9) IBD patients. Taurine can be derived from the diet, and the foods with a high taurine content are seafood and dark meat poultry [16]. When IBD patients were stratified according to their weekly consumption of such foods, we found no significant difference among patients either eating or not seafood or dark meat poultry (Figure 3B). We also evaluated whether the taurine levels were influenced by the weekly intake of taurine-containing energy drinks and, again, found no statistical difference (Figure 3B). 

The serum taurine levels were significantly reduced in IBD patients with clinically active disease compared to those in remission (Figure 4A). However, when patients were stratified according to the IBD type, a significant difference between patients with clinically active disease and those in remission was confirmed in UC but not in CD (Figure 4B,C). Finally, the taurine levels were not influenced by disease extent in UC, and by disease localization and phenotype in CD (Figure 4D–F). 

## 4. Discussion

The present study was undertaken to examine whether in IBD there is a reduction in the serum levels of taurine, a semi-essential micronutrient that reduces cellular senescence, decreases DNA damage, suppresses mitochondrial dysfunction, protects against telomerase deficiency, and attenuates inflammation in many organs, including the gut [15,17]. Our data indicate that the circulating levels of taurine are reduced in both CD and UC compared to the controls, with no significant difference between UC and CD. These findings confirm and expand on the results of previous studies documenting reduced levels of taurine in the blood samples of patients with other illnesses, such as chronic kidney diseases and Parkinson’s disease [18,19]. 

In humans, taurine levels decline with age and correlate negatively with several age-associated diseases, situations that are often related to depression of the immune system [15,20]. In line with this, the levels of taurine in our control population declined with age, and individuals older than 50 years had serum levels of taurine significantly lower than those in individuals with age <50 years. In contrast, in IBD, no decline in taurine levels was documented with age, and patients younger than 50 years had the levels of taurine reduced as compared to age-matched controls, thus excluding the possibility that the taurine deficiency in IBD relies on the relatively older age of our patients (46 years) compared to the controls (35 years). 

To explore the factors involved in IBD-associated taurine deficiency, we compared the levels of taurine with several characteristics of IBD patients and found no significant difference among patients with different extents/localizations of the lesions or CD phenotype. In contrast, patients with clinically active IBD had levels of taurine significantly reduced compared to those of patients in remission. However, when patients were stratified according to the IBD type, the difference was significant for UC but not CD. We are aware that such findings deserve further investigation since the disease activity was evaluated exclusively by clinical scores given the lack of fecal calprotectin and endoscopy data in the majority of the patients. In patients with CD, there is a lack of correlation between clinical scores of activity and the endoscopic appearance of the ileocolonic mucosa, raising the possibility that some of our CD patients could have an active disease despite the absence of clinical symptoms [21]. While our study was ongoing, it was shown that the blood taurine levels were reduced in patients with active CD compared to those in remission, and the levels of taurine before the beginning of therapy were significantly lower in patients who did not achieve clinical remission following anti-TNF treatment [22]. Overall, these data are conflicting with those published by Zhou and colleagues who documented higher levels of taurine in the serum samples of CD patients compared to the controls, while in UC, there was a slight reduction in taurine levels compared to the controls [23]. Interpretations of this discrepancy are tempered by design differences, including the methods adopted to analyze taurine and the lack of information on the IBD activity in Zhou’s study. 

The mechanism underlying the reduction in circulating taurine in IBD remains to be ascertained. In a cohort of 11,966 individuals with aging-associated pathologies from the EPI-Norfolk study, integrating more than 50 clinical risk factors, Pietzner and colleagues observed that increases in taurine levels were associated with a lower BMI and abdominal obesity, and an increase in all taurine-related metabolites was associated with a lower prevalence of type II diabetes, less liver damage, and inflammation [24]. However, our analysis showed that in IBD, the taurine levels did not differ among patients with different values of BMI. Taurine is obtained mainly through diet in humans, with minuscule amounts made endogenously. No difference was found among patients either eating or not foods high in taurine. Similarly, the levels of taurine did not differ between patients with a weekly intake of taurine-containing energy drinks and those with no intake. Nonetheless, we cannot exclude the possibility that our analysis has been confounded by other unmeasured factors related to taurine intake at the individual level. Studies are now ongoing to ascertain whether IBD-associated taurine deficiency is secondary to the mucosal inflammation-driven consumption of the molecule, as well as to assess the gene expression and enzymatic activity of cysteine sulfinic acid decarboxylase in IBD. 

Whatever the regulatory mechanisms operating on endogenous taurine levels, our data support the hypothesis that taurine deficiency could play a role in IBD pathogenesis, given that reduced levels of taurine have been associated with inflammation and anomalous immune phenotypes [25], and dietary supplementation of taurine is efficient in suppressing pathogenic inflammatory responses in the gut [25,26].

In conclusion, our study shows that IBDs are characterized by reduced levels of taurine, which would seem to reflect the activity of the disease, at least in UC. These data support further experimentation aimed at assessing the effect of dietary supplementation with taurine on the disease activity in subgroups of patients with reduced serum levels of the molecule.

## Figures and Tables

**Figure 1 nutrients-16-01593-f001:**
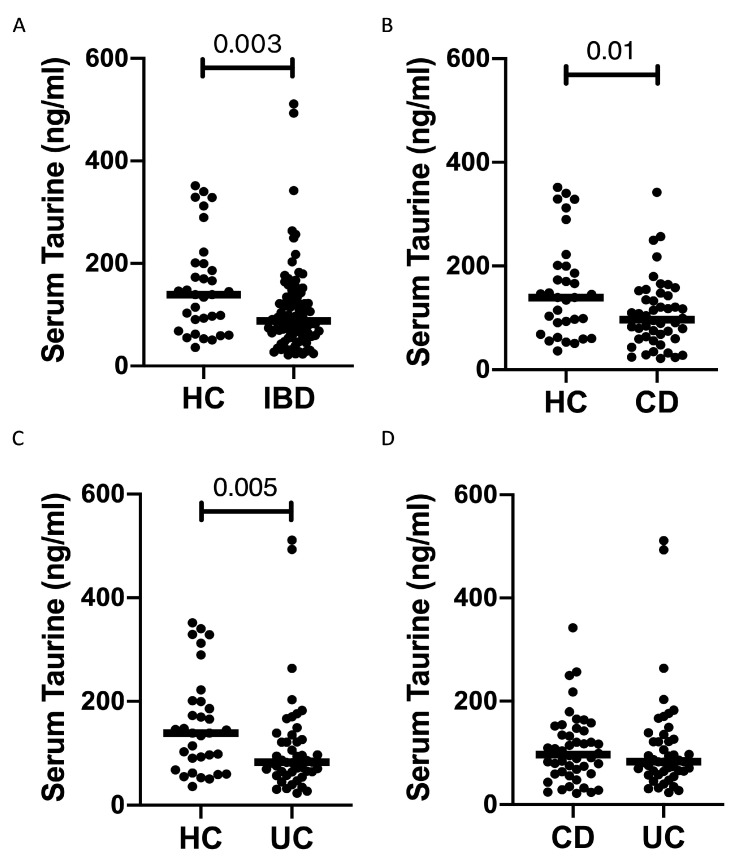
Patients with inflammatory bowel diseases have reduced taurine levels in the serum. (**A**) Taurine was measured in the serum samples of 33 healthy controls (HC) and 92 patients with inflammatory bowel disease (IBD). Each point on the graph represents the taurine level in an individual serum sample, with horizontal bars indicating the median value for each group. (**B**–**D**) Taurine levels were measured in the serum samples of 33 healthy controls (HC), 46 Crohn’s disease (CD) patients (**B**,**D**), and 46 ulcerative colitis (UC) patients (**C**,**D**). Each point on the graph represents the taurine level in an individual serum sample, with horizontal bars indicating the median value for each group.

**Figure 2 nutrients-16-01593-f002:**
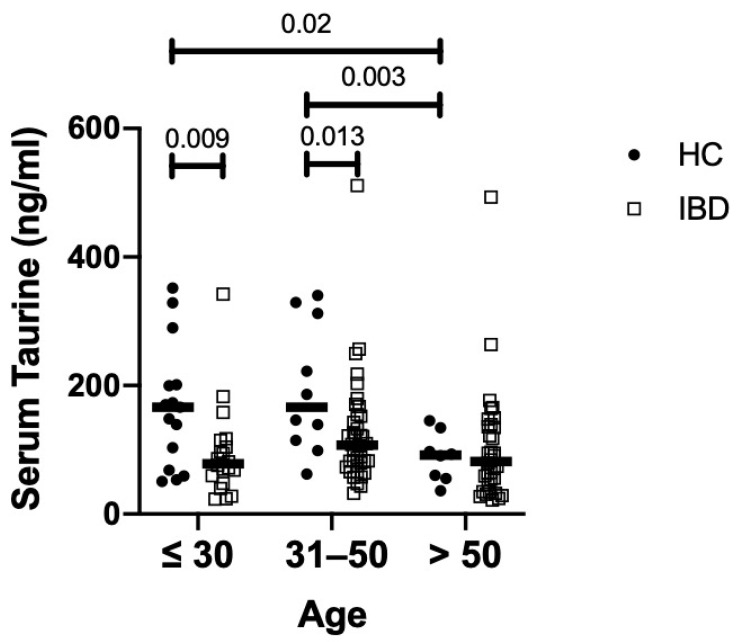
Serum taurine levels in subgroups of IBD patients and healthy controls (HC) according to their age. Each point in the graph indicates the level of taurine in the serum sample of a single patient or control, while the horizontal bars indicate the median value for each group.

**Figure 3 nutrients-16-01593-f003:**
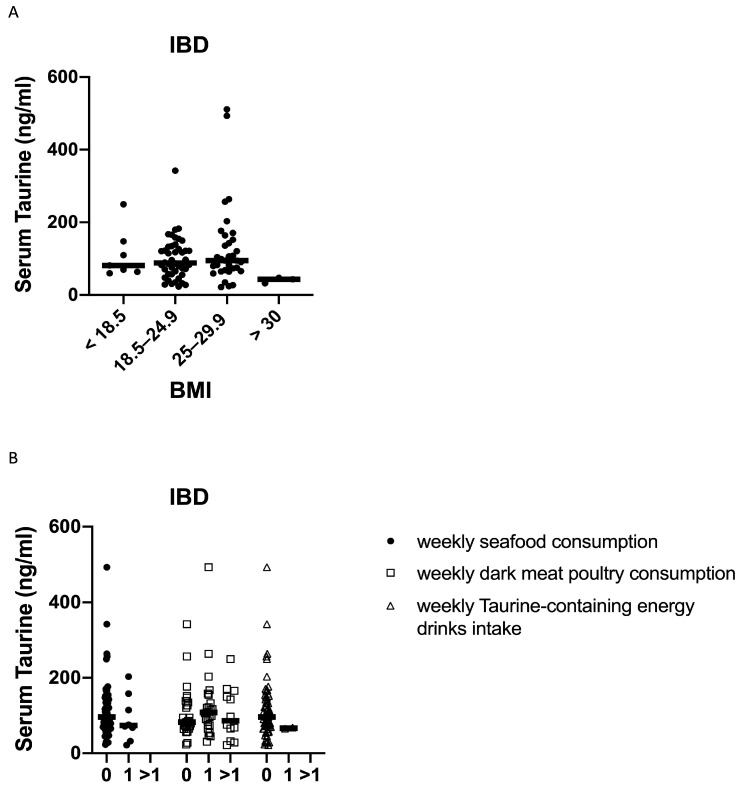
Serum taurine levels in subgroups of IBD patients according to their body mass index (BMI) (**A**), weekly consumption of seafood and dark meat poultry, and weekly intake of taurine-containing energy drinks (0 = 0 times a week, 1 = once a week and >1 = more than once a week) (**B**). Each point in the graph indicates the level of taurine in the serum sample of a single patient, while the horizontal bars indicate the median value for each group.

**Figure 4 nutrients-16-01593-f004:**
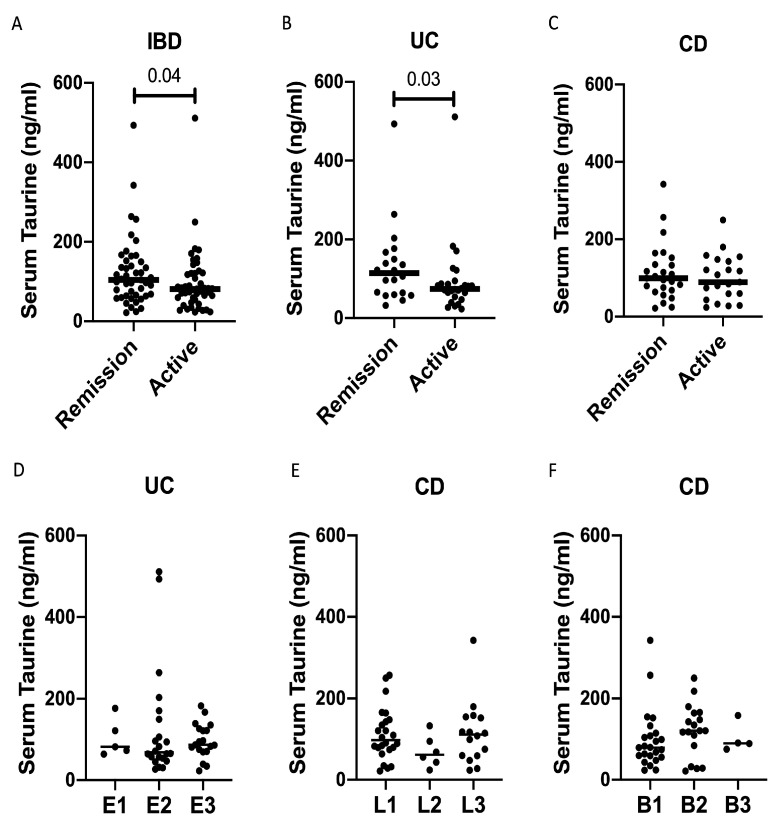
Taurine levels are reduced in the serum samples of patients with active inflammatory bowel diseases (IBDs). (**A**) Taurine was measured in the serum samples of 47 IBD patients with clinically active disease and 45 IBD patients in clinical remission. Each point in the graph indicates the level of taurine in the serum sample of a single patient, while the horizontal bars indicate the median value for each group. (**B**) Taurine was measured in the serum samples of 26 UC patients with clinically active disease and 20 UC patients in clinical remission. Each point in the graph indicates the level of taurine in the serum sample of a single patient, while the horizontal bars indicate the median value for each group. (**C**) Taurine was measured in the serum samples of 21 CD patients with clinically active disease and 25 CD patients in clinical remission. Each point in the graph indicates the level of taurine in the serum sample of a single patient, while the horizontal bars indicate the median value for each group. (**D**–**F**) Taurine was measured in the serum samples of 46 UC patients and 46 CD patients, which were divided taking into account the UC extent (**D**), CD localization (**E**), and phenotype (**F**). Each point in the graph indicates the level of taurine in the serum sample of a single patient, while the horizontal bars indicate the median value for each group.

**Table 1 nutrients-16-01593-t001:** Demographic and clinical characteristics of ulcerative colitis (UC) patients and Crohn’s disease (CD) patients.

Characteristic	UC Patients (*n* = 46)	CD Patients (*n* = 46)	*p*-Value
Age (years), median (range)	46 (20–69)	46 (19–76)	0.832
Female gender, *n* (%)	25 (54.3)	18 (39.1)	0.210
Weekly seafood consumption, *n* (%): 0 1 >1	30 (88.2) 4 (11.8) 0 (0)	25 (86.2) 4 (13.8) 0 (0)	-
Weekly dark meat poultry consumption, *n* (%): 0 1 >1	15 (44.1) 13 (38.2) 6 (17.6)	11 (37.9) 12 (41.4) 6 (20.7)	-
Weekly taurine-containing energy drinks, *n* (%): 0 1 >1	33 (97) 1 (3) 0 (0)	28 (96.5) 1 (3.5) 0 (0)	-
IBD patients with clinically active disease, *n* (%)	26 (56.5)	21 (45.7)	0.400
Therapy with mesalamine, *n* (%)	29 (63)	15 (32.6)	0.007
Therapy with steroids, *n* (%)	4 (8.7)	8 (17.4)	0.353
Therapy with biologic agents, *n* (%)	8 (17.4)	19 (41.3)	0.022
Therapy with immunosuppressants, *n* (%)	0 (0)	5 (10.9)	-
UC localization, *n* (%): E1 E2 E3	5 (10.9) 22 (47.8) 19 (41.3)	-	-
CD localization, *n* (%): L1 L2 L3	-	24 (52.2) 6 (13) 16 (34.8)	-
CD phenotype, *n* (%): B1 B2 B3	-	24 (52.2) 18 (39.1) 4 (8.7)	-

## Data Availability

The original contributions presented in the study are included in the article, further inquiries can be directed to the corresponding author/s.

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
