# Peer review of "Reduced Taurine Serum Levels in Inflammatory Bowel Disease"

_nutrients, 2024, doi:10.3390/nu16111593_

Round 1
Reviewer 1 Report
Comments and Suggestions for Authors
1. Does the taurine decrease in IBD in common in people from other regions such as Asia?
2. Is there a difference between men and women? Is there a difference according to gender?
3. Can you say that eating foods high in Taurine prevents IBD?
4.This paper is of great significance in that it was conducted on humans as a paper confirming the taurine concentration with ELISA in humans.
5. How can the relationship between Taurine reduction and IBD be explained?
Author Response
We would like to thank the reviewer for his/her positive evaluation and helpful comments/suggestions. In response to the specific issues raised by this reviewer:
- Does the taurine decrease in IBD in common in people from other regions such as Asia?
Response: The reviewer has raised an important point, as geographical differences may have an impact. Unfortunately, we do not have any data in this regard, as our study was conducted in Italy, in a Caucasian population. Further studies from other groups/countries will likely answer this question.
- Is there a difference between men and women? Is there a difference according to gender?
Response: We would like to thank the reviewer for her/his helpful suggestion. We have performed an additional analysis and found no difference in taurine levels according to gender. These data have been added to the revised version of the manuscript.
- Can you say that eating foods high in Taurine prevents IBD?
Response: This is a good point but the current available data are not sufficient to conclude that foods high in taurine can prevent IBD. Nonetheless, data from preclinical models of IBD (as discussed in the manuscript) can attenuate IBD-like colitides.
4.This paper is of great significance in that it was conducted on humans as a paper confirming the taurine concentration with ELISA in humans.
Response: We would like to thank the reviewer for her/his positive evaluation.
- How can the relationship between Taurine reduction and IBD be explained?
Response: The mechanisms underlying the reduction of circulating taurine in IBD remains to be ascertained. We found no correlation between taurine levels and age, food consumption or BMI. In patients with active ulcerative colitis, but not with Crohn disease, taurine levels were decreased. Further investigations in IBD cohorts are needed to assess whether taurine reduction is due to enhanced consumption. These points were discussed in the manuscript.
Reviewer 2 Report
Comments and Suggestions for Authors
This enlightening article shows the role of the often-overlooked non-proteinogenic amino acid taurine in IBD. This prospective study is succinct, but straightforward. I have some remarks:
Results: Although differences are statistically significant, the variance in both groups is large, with many overlapping values in both of them. Could relative values over time be more important than the absolute values at any given time? There is also a small group of outliers: is this a sub-population?
Discussion: Correlation does not necessarily mean causality. Could the taurine levels in elderly patients be due to the length of their disease, rather than its severity?
Author Response
We would like to thank the reviewer for his/her positive evaluation and helpful comments/suggestions. In response to the specific issues raised by this reviewer:
This enlightening article shows the role of the often-overlooked non-proteinogenic amino acid taurine in IBD. This prospective study is succinct, but straightforward. I have some remarks:
Results: Although differences are statistically significant, the variance in both groups is large, with many overlapping values in both of them. Could relative values over time be more important than the absolute values at any given time? There is also a small group of outliers: is this a sub-population?
Response: We agree with the reviewer that it would be interesting to monitor taurine levels in the same patients over time. This longitudinal study would take a time much longer than that needed to re-submit our revised work. However, we feel that follow-up studies will be needed to better clarify the relevance of taurine reduction in IBD and further assess the factors involved in such decrease. We analyzed the group of outliers and found no disease or demographic characteristics that distinguished them from the others.
Discussion: Correlation does not necessarily mean causality. Could the taurine levels in elderly patients be due to the length of their disease, rather than its severity?
Response: We feel this is not the case because, in our IBD population, taurine levels did not significantly change according to age and the levels of taurine in patients older than 50 years were similar to those seen in patients younger than 50 years. In contrast, the reduction of taurine level was seen only in healthy individuals >50 years old.
Round 2
Reviewer 1 Report
Comments and Suggestions for Authors
It is well revised.